# Is addressing violence against women prioritised in health policies? Findings from a WHO policies database

**Eva Burke**[1]*, **Katherine Watson**[2], **Gillian Eva**[3], **Judy Gold**[4], **Claudia Garcia-Moreno**[5], **Avni Amin**[5]

1 Freelance Consultant, Bath, United Kingdom, 2 Freelance Consultant, Torchlight Collective, Nashville, Tennessee, United States of America, 3 Freelance Consultant, Washington, DC, United States of America, 4 Freelance Consultant, Cultivating Change, Melbourne, Australia, 5 Department of Sexual and Reproductive Health and Research, World Health Organization, Geneve, Switzerland

* eva.burke.consulting@gmail.com

**Data Availability Statement:** Data reflect WHO Member State information and are not yet publicly available. Researchers will be able to obtain the

## Abstract

Violence against women (VAW) affects almost 1 in 3 women and can lead to short and long-term adverse health outcomes. The health sector is an important entry point to respond to VAW. Globally, countries have committed to eliminating VAW through the SDGs and WHO Member States have endorsed a Global Plan of Action on Violence, which asks countries to provide comprehensive health services to VAW survivors. To track progress and establish a baseline for the Global Plan of Action on Violence, WHO developed a VAW Policies Database to assess how countries are addressing VAW in health and multisectoral policies. This paper presents findings from 15 select indicators related to the existence of VAW-related policies and the inclusion of health services for survivors in policies in line with WHO recommendations. Results show that while 80% of countries have multisectoral VAW policies in place, only 34% have national health policies that include VAW response and/or prevention as a strategic priority, and 48% have clinical guidelines for the health sector response. Policies were analysed to identify inclusion of WHO-recommended VAW health services: 75% of countries' policies mention provision of first-line support; while 50% or fewer of countries mention clinical enquiry to identify intimate partner violence, post-rape care services, or mental health assessment, referral and treatment. The high-income countries group had the lowest proportion of countries that specified each of the above-mentioned health services in policies. Findings show that more efforts are needed to raise the awareness of ministries of health about the public health impact of VAW and the importance of including VAW in health policies. Where policies exist, many are not aligned with WHO-recommendations. Policy dialogues should be conducted with governments on how to better align their VAW policies with internationally-agreed, evidence-based standards, and to implement them through programmes and services.

data set by requesting it from the WHO secretariat at srhavp@who.int. In the future, we are striving to make the data available through the WHO's SRH policy portal: https://platform.who.int/data/sexual-and-reproductive-health-and-rights/national-policies.

**Funding:** Funding for this work was provided by a grant to WHO through the Government of UK (Foreign, Commonwealth and Development Office – FCDO) and by the UNDP/UNFPA/UNICEF/World Bank/WHO Special Programme of Research on Human Reproduction (HRP). The funders had no role in study design, data collection and analysis, decision to publish, or preparation of the manuscript. Initials of author who received the reward: AA.

**Competing interests:** The authors have declared that no competing interests exist.

## Introduction

Violence against women (VAW) is defined by the United Nations (UN) as "*any act of gender-based violence that results in, or is likely to result in, physical, sexual or mental harm or suffering to women, including threats of such acts, coercion or arbitrary deprivation of liberty, whether occurring in public or in private life*" and can occur in the family, in the community, or be perpetrated or condoned by the State [1]. The World Health Organization's (WHO) 2018 estimates of VAW prevalence show that nearly 1 in 3 women have experienced physical and/or sexual violence from a male intimate partner and/or sexual violence from someone other than an intimate partner at least once in their lifetime since the age of 15 [2]. The same study also highlights that intimate partner violence (IPV) (whether physical, sexual or psychological) is the most widespread form of violence against women globally, with an estimated 27% of women aged 15–49 years reporting experience of physical and/or sexual violence by an intimate partner at least once in their lifetime and 13% in the last 12 months [2]. IPV prevalence in the last 12 months was highest in Oceania, Southern Asia and Africa. An estimated 6% of women aged 15–49 years have experienced non-partner sexual violence (NPSV) from someone other than a current or former husband or male intimate partner at least once since the age of 15 [2]. Reported incidents of violence against women have increased in some settings since the outbreak of COVID-19, coinciding with the reduction of VAW services in many countries due to pandemic-era restrictions [3].

The health impact of VAW is well-recognised and documented. VAW can lead to short- and long-term adverse physical and mental health outcomes. Physical outcomes include sexual and reproductive health problems such as unintended pregnancies, adverse maternal and newborn health outcomes, sexually transmitted infections (STIs) and HIV infection, gynaecological problems, and injuries. Mental health outcomes include depression, post-traumatic stress disorder (PTSD), and suicide [2]. Women who experience violence are more likely to seek health services, even if they do not explicitly disclose violence to a health care provider [4, 5]. Therefore, the health sector is a critical entry point for addressing VAW within a multisectoral response.

Over the past decade, governments have committed in both international and regional agreements to prevent and address VAW. This includes several UN General Assembly resolutions (UN, 1993), regional conventions such as the Inter-American Convention of Belem do Para [6] and the Council of Europe Convention on preventing and combating VAW and domestic violence (Istanbul Convention) [7]. The 2030 Sustainable Development Goals (SDG) specifically include a target and indicators related to VAW under SDG 5 for gender equality. Target 5.2 focuses on the elimination of all forms of violence against all women and girls in the public and private spheres and is tracked by indicators 5.2.1 and 5.2.2 related to prevalence of IPV and sexual violence in the last 12 months, respectively [8]. In recognition of the public health burden of VAW, at the 69th World Health Assembly in 2016, WHO Member States endorsed resolution WHA69.5 –the *Global Plan of Action to Strengthen the Role of the Health System within a National Multisectoral Response to Address Interpersonal Violence, in particular Against Women and Girls, and Against Children* (henceforth '*the Global Plan of Action on Violence*'). The Global Plan of Action on Violence calls on WHO Member States to: explicitly address VAW in health policies, clinical protocols and guidelines in line with WHO recommendations and international human rights standards; train health care providers and provide comprehensive health services for survivors including sexual and reproductive and mental health care; foster prevention programmes by ensuring that the health sector is reflected in the multisectoral VAW plans and is working with other sectors to promote evidence-based prevention; and strengthen research, evidence and data including through prevalence surveys and inclusion of VAW response indicators in health surveillance [9].

In 2020, WHO commissioned the development of a VAW Policies Database, aiming to assess how, if at all, countries are addressing VAW in health and multisectoral policies, including whether their content is aligned with WHO recommendations, evidence-based strategies, and international human rights standards. The VAW Policies Database also provides a baseline against which to monitor progress of the Global Plan of Action on Violence and to facilitate policy dialogue with governments. This paper presents select findings in relation to two groups of indicators in the VAW Policies Database, presented by SDG geographical regions and World Bank country income groups: 1) the existence of health and multisectoral policies that address VAW and 2) the extent to which health services recommended by WHO to address VAW are included within such policies. The findings highlight what actions need to be taken by governments to strengthen their policy frameworks for improving the health sector's response to VAW going forward.

## Methods

The VAW Policies Database includes data for 54 indicators covering six areas of measurement: 1) existence of policies addressing VAW; 2) inclusion of health services recommended by WHO in policies; 3) availability of services recommended by WHO; 4) extent to which policies address groups living in situations of vulnerability; 5) inclusion of evidence-based prevention strategies in policies; and 6) availability of prevalence data on VAW. Of these, only findings reflecting the first two areas of measurement covering 15 indicators are presented in this paper. The VAW Policies Database is populated with evidence from the content analysis of over 600 health and multisectoral policy documents. The three main types of policy documents that were reviewed for the VAW Policies Database are outlined in Table 1.

We refer to all policy documents that were included in the VAW Policies Database collectively in this paper as 'VAW policies'. All included policy documents had to be government-endorsed and final versions of national-level policies. Sub-national policies were not included, nor were documents issued by third-party institutions. Policy documents were sourced from existing repositories, including the global policy repository compiled from the 155 country responses to the 2018–2019 WHO sexual, reproductive, maternal, newborn, child and adolescent health (SRMNCAH) policy survey, as well as those managed by regional and country WHO offices. Extensive online searches were also conducted to source policy documents. In addition, the WHO office in the Americas did targeted outreach to countries to request policy documents for the VAW Policies Database. The findings presented in this paper are based on data included in the VAW Policies Database as of November 2021.

**Table 1. The main types of policy documents included in the VAW policies database.**

| Policy type | Inclusion criteria |
|---|---|
| **National health policy** | Any national health sector-specific policy document that addresses VAW, or including general, sexual and reproductive health (SRH), HIV or reproductive, maternal, newborn, child and adolescent health (RMNCAH) health policies, that addresses VAW. Adolescent and maternal policies, emergency preparedness and mental health policies were not included. |
| **National multisectoral VAW policy** | National VAW multisectoral policy document or gender equality and/or other policies that address the advancement of women and contain a strong VAW component. |
| **Clinical guidelines** | Guidance for the health sector's prevention and/or response to VAW. Although the broad term 'clinical guidelines' has been used for this category, it can refer to any national health sector-specific document such as clinical guidelines, standard operating procedures (SOPs) or guidance for health-care providers, managers and/ or administrators to respond to VAW. |

A global status report published by WHO on addressing VAW in health and multisectoral policies includes more details of the data sources, methodology, indicators and findings from all 54 indicators in aggregate at the global level and disaggregated by WHO regions [10]. In contrast, the findings herein are presented by SDG super regions (hereafter referred to as 'SDG regions'): Africa, Americas, Asia, Europe, Oceania [11] and the four World Bank income groups: low-income, lower-middle income, upper-middle income, and high-income [12]. For the indicators related to the existence of VAW-related policies, additional analysis was done by the 17 SDG sub-regions, which are: Australia and New Zealand; Central Asia; Eastern Asia; Eastern Europe; Latin America and the Caribbean; Melanesia; Micronesia; Northern Africa; North America; Northern Europe; Polynesia; South-eastern Asia; Southern Asia; Southern Europe; Sub-Saharan Africa; Western Asia; Western Europe. A subset of 15 indicators were prioritised for this paper due to their importance to the health sector's response to VAW; these are topics that can reasonably be expected to be found in the types of health and multisectoral policies included in the VAW Policies Database. All WHO Member States (n = 194) were considered in the denominator for indicators related to the existence of health policies, but for results presented by World Bank income regions, however, there are two fewer countries given that the Cook Islands and Niue do not have World Bank income categorisation (n = 192). The denominator for inclusion of health service in policies indicators is based on only those countries for which policy documents were available and therefore, uploaded to the VAW Policies Database (n = 174), but for results presented by World Bank income regions, however, there is one fewer country given that the Cook Islands does not have a World Bank income categorisation (n = 173).

## Results

The Global Plan of Action on Violence suggests that countries include VAW prevention and response interventions in health policies and develop guidelines, protocols or SOPs to guide health providers [9]. Accordingly, the first group of indicators for which findings are highlighted below relate to the existence of these policies and protocols (Table 2).

WHO guidelines [13] provide detailed recommendations for the different types of health services that should be available for survivors of VAW (see S1 Table). Given these recommendations, findings for the second grouping of indicators highlight data on the extent to which policies include: methods for the identification of IPV; provision of first-line support; provision of post-rape care, and; provision of mental health services (i.e. assessment, referral and treatment) (Table 3).

### Existence of health and multisectoral policies that include VAW

National health policies were assessed for whether they included VAW response and/or prevention as a strategic priority, which was defined as VAW being one of the primary goals, objectives or strategic priorities articulated within the policy. Globally, 34% of countries have national health policies that include VAW response and/or prevention as a strategic priority (Fig 1).

**Table 2. Indicators related to the existence of VAW-related policies.**

| Existence of VAW-related policies | Proportion of countries with a national health policy that includes VAW response and/or prevention as a strategic priority, a multisectoral VAW policy that includes the health sector), or clinical guidelines, by SDG regions and sub-regions |
|---|---|
| | Proportion of countries with a national health policy that includes VAW response and/or prevention as a strategic priority, a multisectoral VAW policy that includes the health sector, or clinical guidelines, by World Bank income group |

**Table 3. Indicators related to the inclusion of VAW health services in policy.**

| Existence of VAW health services in policy | Proportion of countries that include clinical enquiry or universal screening in policy, by SDG regions and World Bank income groups |
| --- | --- |
| | Proportion of countries that include first line support in policy, by SDG regions and World Bank income groups |
| | Proportion of countries that include EC, HIV PEP, STI treatment, and all three services, in policy, by SDG regions and World Bank income groups |
| | Proportion of countries that include abortion for survivors of VAW in policy, by SDG regions and WB income groups |
| | Proportion of countries that include mental health assessment and referral in policy, by SDG region and World Bank income groups |
| | Proportion of countries that include mental health treatment in policy, by SDG regions and World Bank income groups |

The SDG region of Oceania has the highest proportion of countries (63%) with national policies that include VAW response and/or prevention as a strategic priority, followed by the Americas (57%). On a SDG sub-regional level, there are wide disparities; more than half of countries in the sub-regions of Australia and New Zealand, Melanesia, Polynesia, Latin America and the Caribbean, and Southern Asia have national health policies that include VAW response and/or prevention as a strategic priority (see S2 Table). By contrast, no countries in the sub-regions of Central Asia, East Asia and Northern Africa have VAW response and/or prevention as a strategic priority within national health policies. Looking at data by income

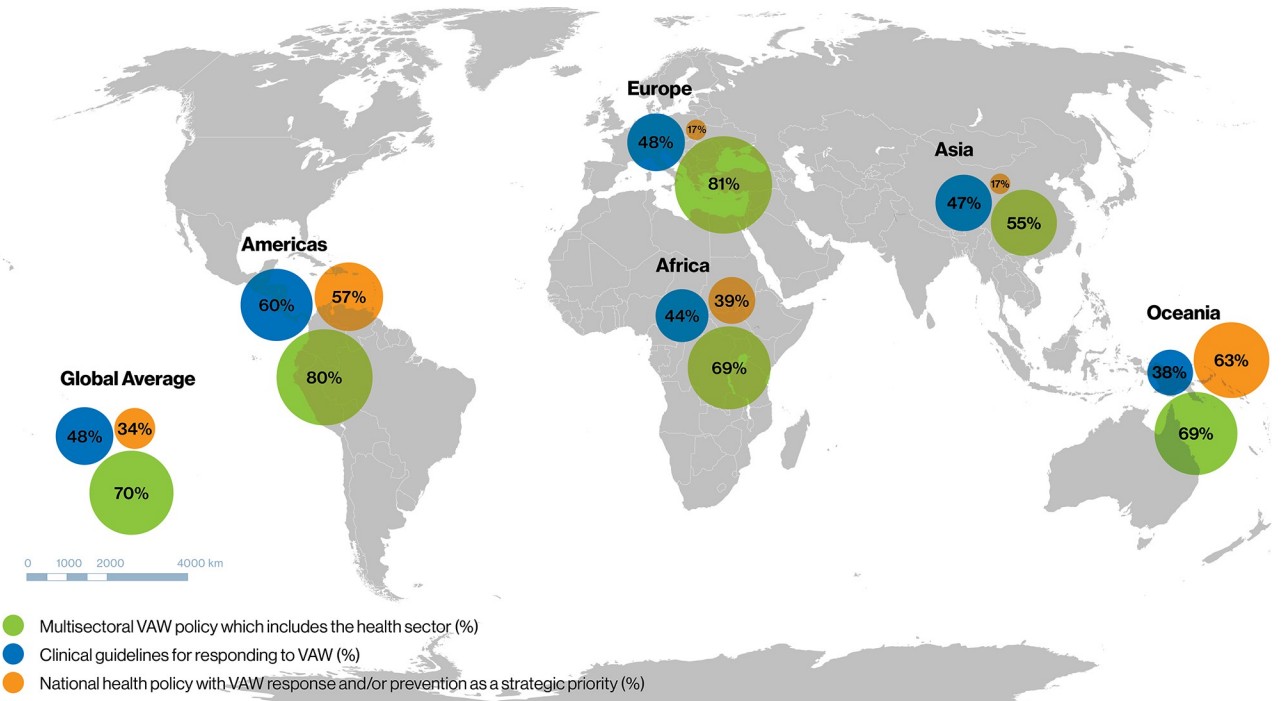

**Fig 1. Existence of health sector VAW-related policies, by SDG regions.** Map base layer source: https://en.m.wikipedia.org/wiki/File:BlankMap-World.svg i. Note that n = 194 for SDG regions but n = 192 for World Bank income groups because two countries (Cook Islands and Niue) are not assigned to a World Bank income group.ii. Global averages for SDGs are included here (n = 194). The global averages (%) are the same for both groupings (SDG and World Bank). iii. The designations employed and the presentation of the material in this publication do not imply the expression of any opinion whatsoever on the part of WHO concerning the legal status of any country, territory, city or area or of its authorities, or concerning the delimitation of its frontiers or boundaries. Dotted and dashed lines on maps represent approximate border lines for which there may not yet be full agreement.

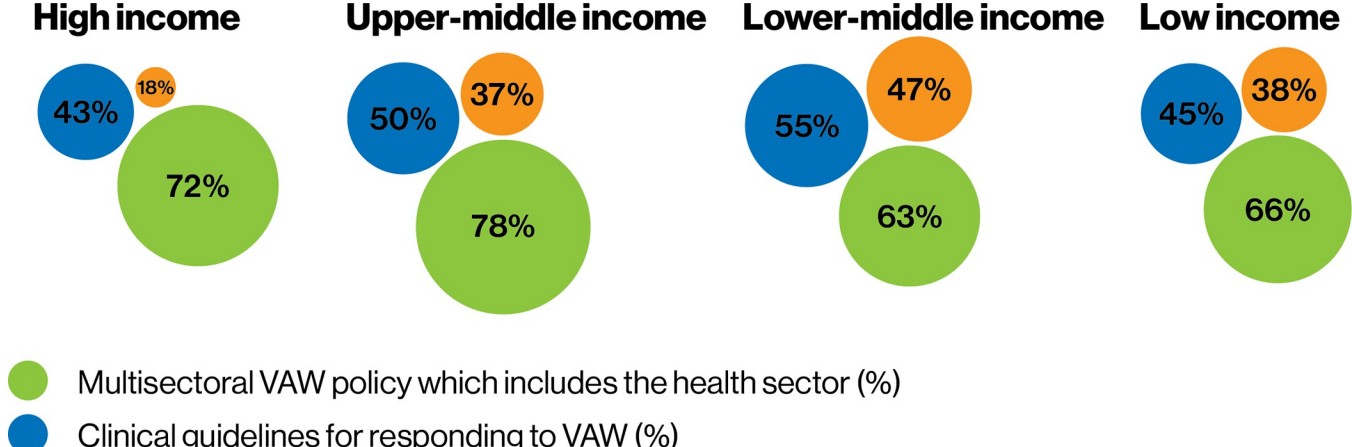

**Fig 2. Existence of health sector VAW-related policies, by World Bank income groups.**

groups, the high-income group has the lowest proportion of countries with national health policies that include VAW response and/or prevention as a strategic priority (Fig 2) (S3 Table).

The Global Plan of Action on Violence emphasises the need for a multisectoral response that includes the health sector. While a majority of countries (81%) have multisectoral VAW plans or policies (S2 Table), slightly fewer (70%) have multisectoral VAW plans or policies that explicitly mention the health sector role or response to VAW (Fig 1). In most SDG regions, between 12% and 14% of multisectoral VAW plans do not include the health sector role in VAW, with the exception of the Americas where all multisectoral VAW plans include the health sector. The proportion of countries with a multisectoral VAW plan that explicitly includes the health sector also varies by SDG sub-region: most or all countries in Australia and New Zealand and Eastern, Northern and Western Europe have multisectoral VAW plans that include health sector, but under half of countries in Central Asia and Southern Asia include it (S2 Table).

Multisectoral VAW plans that include the role of the health sector are found for approximately three quarters of upper-middle and high-income countries and approximately two thirds of lower-middle and low-income countries (Fig 2).

Nearly half (48%) of countries have clinical guidelines for responding to VAW, ranging from the highest proportion of countries in the SDG region of the Americas (60%) to the lowest in Oceania (38%) (Fig 1). Analysis by SDG sub-regions shows more stark differences, with no countries with clinical guidelines for responding to VAW in Eastern Asia, North America, and Polynesia, whereas most or all of the countries in the SDG sub-regions of Melanesia, Southern Asia, and Australia and New Zealand have clinical guidelines (S2 Table). Although there are not major variances between the different income groups, a slightly higher proportion of lower-middle income countries have clinical guidelines responding to VAW than countries from other income groups (Fig 2).

Looking across the different types of policies, multisectoral VAW plans are the most commonly found types of policy across all SDG regions and income groups. Across SDG regions and income groups, there is some variation of the proportion of countries with clinical guidelines, but variation is more noticeable between SDG regions, sub-regions and income groups in the proportion of countries with health policies that include VAW as a strategic priority.

## Inclusion of health services for survivors in policies in line with WHO recommendations

Approximately a quarter (24%) of countries have policies that align with WHO's recommendation to use clinical enquiry to identify women subjected to IPV. There are not significant differences in this indicator between SDG regions, though Asia has the highest proportion of countries with policies that include clinical enquiry (S4 Table). Analysing by income group, the highest proportion of countries that include clinical enquiry in their VAW policies are in the lower-middle income group, whilst the lowest proportion is in the high-income group of countries (S4 Table).

The provision of first line support and its inclusion in VAW policies was considered to be met if a policy document mentioned the importance of one or more of the following elements: a) providing practical care and support; b) listening; c) offering validation or comfort; and d) offering information about or connecting survivors of VAW to other support services. Three-quarters (75%) of countries include first line support for survivors in policy, although this is higher among countries in the SDG regions of the Americas and Africa and lowest in Europe (S5 Table). More countries from the low and lower-middle income group include first line support in policy than in high and upper-middle income countries (S5 Table).

Just over half of countries include information about the provision of each of the following three components of post-rape care—emergency contraception (EC), HIV post-exposure prophylaxis (PEP) and STI prophylaxis—in policies, whilst just under half of all countries include *all three* services (Table 4). A higher proportion of countries in the Americas and Oceania regions include *all three* services, as compared with other SDG regions. Europe has the lowest proportion of countries with policies that include each of the post-rape care services and the lowest proportion of countries with policies that include *all three* services in policy. By income group, similar proportions of low-income, lower middle-income and upper-middle countries include each of the three post-rape care elements and *all three* services, but fewer high-income countries include *all three* in policy when compared to other income groups (Table 4).

The inclusion of abortion as an element of post-rape care in VAW policies is reported separately because a number of countries have laws against the provision of abortion on different

**Table 4. Proportion of countries that include EC, HIV PEP, STI treatment, and all three services, in policy, by SDG region and World Bank income group.**

|  | EC (%) | HIV PEP (%) | STI treatment (%) | EC, HIVPEP, and STI treatment (%) |
|---|---|---|---|---|
| **SDG region** | | | | |
| Africa (n = 50) | 62 | 58 | 50 | 46 |
| Americas (n = 34) | 76 | 79 | 71 | 68 |
| Asia (n = 36) | 53 | 47 | 50 | 47 |
| Europe (n = 41) | 29 | 29 | 37 | 22 |
| Oceania (n = 13) | 69 | 69 | 54 | 54 |
| **World Bank income group** | | | | |
| Low income (n = 25) | 33 | 33 | 37 | 26 |
| Lower-middle income (n = 45) | 68 | 68 | 56 | 56 |
| Upper-middle income (n = 49) | 64 | 58 | 56 | 51 |
| High income (n = 54) | 67 | 67 | 61 | 57 |
| **Global (n = 174)\*** | **56** | **54** | **51** | **45\*** |

\*n = 173 for World Bank income groups due to some countries not classified into income groups by the WB. Due to the different denominator, one difference is observed in the global averages: the % of countries with EC, HIVPEP, and STI treatment is 46%. All other global averages are the same.

Note: The proportions in Table 4 are not mutually exclusive i.e. a country could appear in more than one category for individual services. A country that offers EC, HIV PEP and STI treatment will appear in the individual categories as well as in the 'EC, HIV PEP, and STI treatment' category.

grounds. More detailed information about countries abortion laws and policies, including on grounds of rape are available through the WHO Global Abortion Policies Database (GAPD) [14]. Just 17% of countries include information on abortion as an element of post-rape care in their VAW policies (S6 Table).

Mental health care has been identified as an important intervention for responding to VAW. However, in recognition of the scarcity of specialised mental health professionals at the primary health care level and other settings where VAW services are provided, the recommendations of WHO are: to offer first line support to every survivor who has persistent symptoms related to mental health; to offer basic psychosocial support; to assess for moderate to severe depression and/or post-traumatic stress disorder; and to either offer treatment for these two conditions where specialists are available or refer for specialised care where available and as needed. Therefore, for mental health, data are presented for two indicators: 1) assessment and referral for mental health conditions; and 2) treatment by specialists.

A third of countries include mental health assessment and referral in VAW policy, and this is highest in countries in the Americas and Asia and lowest in countries from Europe (Fig 3). Across all income groups, approximately 40% of countries include mental health assessment and referral in VAW policy, with the exception of high-income countries, where this is significantly lower (22%) (Fig 4) (S7 Table).

Just over half of countries include mental health treatment in VAW policy, and this is highest among countries in the Americas and lowest in Oceania and Europe (S8 Table). There is less variation when results are analysed by income group, with the exception of policies from high-income countries where mental health treatment is less frequently included (S8 Table).

Looking across all VAW health services, the Americas have the highest proportion of countries that include the different health services for survivors in policies, and Europe has the lowest (note that for mental health care, this refers to countries having both mental health assessment and referral, and not just one of these). Across every VAW health service indicator

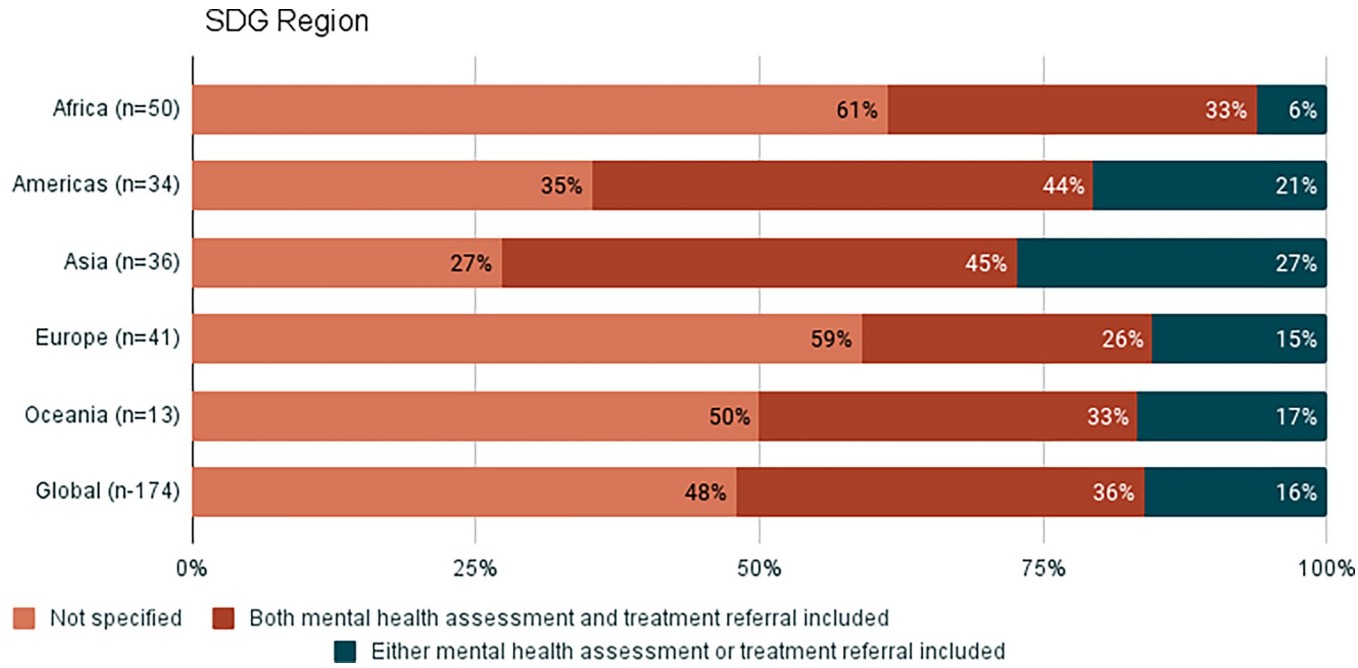

**Fig 3. Proportion of countries that include mental health assessment and referral in policy, by SDG regions.** Categories 'other' and 'unknown' have been removed from the charts to make them easier to read. This data can be found in the supplementary materials, S7 Table.

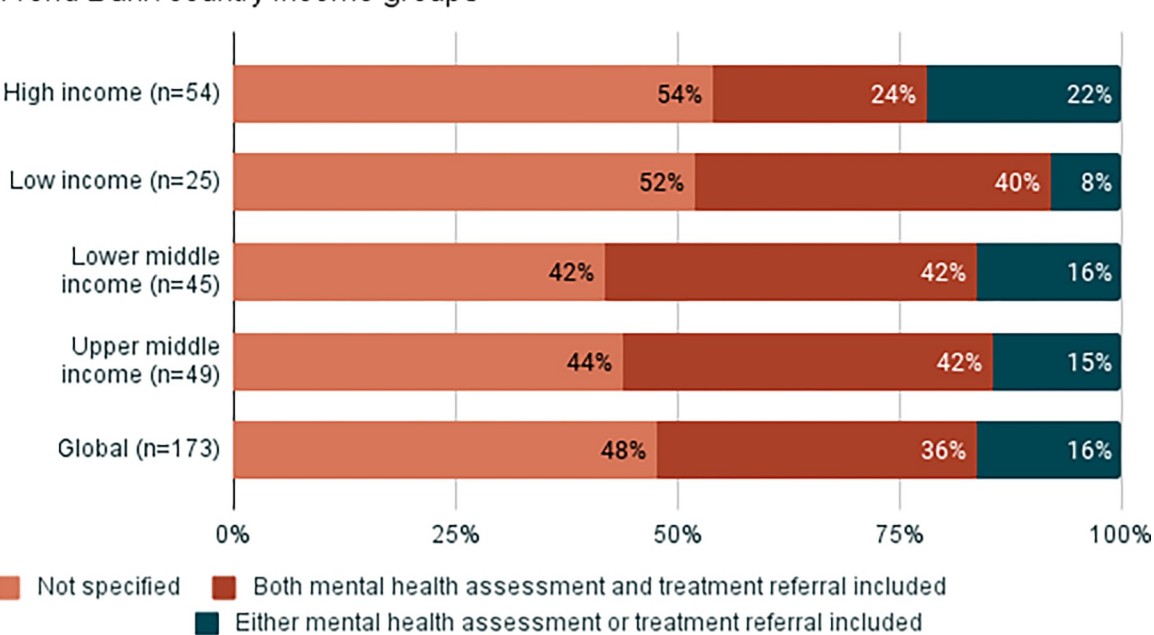

**Fig 4. Proportion of countries that include mental health assessment and referral in policy, by World Bank income groups.** Categories 'other' and 'unknown' have been removed from the charts to make them easier to read. This data can be found in the supplementary materials, S7 Table.

analysed (with the exception of abortion), the high-income group has the lowest proportion of countries that include each health service in VAW policy.

## Discussion

The findings of this paper provide insights into which VAW policies are in place at an aggregate level, and the extent to which they are aligned with evidence-based recommendations as well as international human rights standards. Whilst the need for collaboration across sectors in the prevention and response to VAW is reflected in multisectoral policies for a majority of countries, the same cannot be said of the health sector response, for which fewer national health policies and clinical guidelines for VAW exist. This suggests that there may be variability in whether and how ministries of health perceive VAW as a public health issue and, in turn, the need for awareness raising with ministries of health to illustrate the public health impact of VAW and the crucial role that the health sector plays in prevention and response.

Across SDG regions and income groups, the inclusion of VAW response and/or prevention in health policies varies widely, with fewer high-income countries having national health policies or clinical guidelines for addressing VAW, or including WHO-aligned services in VAW policy. There are several possible explanations for this finding. Firstly, despite best efforts to obtain documents, fewer policy documents were found for countries in the European region—the majority of which are high-income—in comparison to other regions. Secondly, the focus was on obtaining national health policies and multisectoral plans, whilst other documents such as laws, regulations, ministerial orders or sub-national policy documents may be more likely to contain content relevant to VAW in some contexts. A third possible explanation is that in high-income countries, addressing VAW is (rightly) seen as a multisectoral issue that other ministries, such as those related to social development or gender equality, are leading.

Where this is the case, information about VAW prevention and response may be found in policy documents that fell outside of the scope of the VAW Policies Database development, which largely (but not exclusively) focused on the health sector. Lastly, it is also possible that UN agencies including WHO have made more intentional efforts to support ministries of health from low and middle-income countries in the last five to 10 years to address VAW, and this is now being reflected in higher proportions of these supported countries with the requisite policies in place.

Mirroring the variability in whether the health sector response to VAW is reflected in policies, there is also variability in how such policies are addressing WHO standards and international human rights standards in relation to provision of VAW services. WHO recommendations, such as those around the use of clinical enquiry for IPV, and mental health assessment and referral for survivors, are rarely mentioned in VAW policies; this is an area that is ripe for policy dialogues within and between countries. Similarly, abortion services are the least likely component of post-rape care to be included in the VAW policies of most countries, representing an another important area for further policy advocacy.

## Limitations

Like with any policy analysis, there are several limitations to caveat the interpretation of the results. While the main limitations are described in some detail in a separate paper reflecting on the methods used to develop the VAW Policies Database [15] as well as the report published by WHO [10], of particular relevance to this paper, is the possibility that relevant policies were simply not found. While every effort was made to obtain the most up-to-date policies to use for data extraction, it is likely that some policy documents that exist were missed out. The targeted outreach by the WHO office in the Americas to source documents for the VAW Policy Database may have contributed to the Americas region having a higher proportion of countries for which all types of policies were found. In addition, in some countries where changes in government may have recently occurred, the policies at hand may no longer be in use or considered valid.

Finally, caution must be taken when interpreting some of the proportions in the results. The proportions of countries were chosen to allow readers to more easily assess where indicators were, or were not, being met, as well as allowing some comparability across regions and groups. When categorising countries globally into five SDG super regions, 17 SDG subregions, or four World Bank income groups, the denominators can vary substantially. For example, there are just 2 countries in the North America SDG sub-region (so one country is 50% of the total) compared to 48 in the sub-Saharan African SDG sub-region (where 50% of the total is 24 countries).

## Conclusion

Through this global policy analysis we were able to assess the inclusion of VAW in a range of health and other policy documents: the first such analysis of its kind. The resulting information highlights where countries and regions are aligning with the Global Plan of Action on Violence, and opportunities for improvement.

The analysis highlights three issues to address moving forward. First, there is a need to continue with advocacy for ministries of health to engage in, and take greater ownership for, addressing VAW as a public health issue. This is particularly the case for high-income countries and those from the European region where fewer VAW policies were found, but this may also require further research and analysis of how these countries and regions are addressing VAW in their respective policy frameworks. Second, there is a greater need for ministries of

health to develop clinical guidelines to ensure that providers have the necessary guidance to provide services to survivors. Ministries also need to ensure that health policies are intentionally including VAW prevention and response as one of the strategic priorities with budgetary allocations. And finally, greater effort needs to be made to ensure that the content of what is included in policies is better aligned with WHO's evidence-based recommendations, particularly with respect to health response to IPV and the provision of mental health care.

The findings in this paper are limited to the existence of policies and their content in relation to VAW health services and not their status of implementation, nor the budgets allocated to translate policies into implementation, both areas that require further exploration. They do, however, offer a strong starting point for meaningful policy dialogue with governments on how to better align their policies on VAW with agreed standards as well as how to secure the financing and implementation of policies through programmes and services.

## Supporting information

**S1 Table. WHO-recommended services for survivors of VAW (WHO, 2013).**
(DOCX)

**S2 Table. Proportion of countries with a national health policy that includes VAW response and/or prevention as a strategic priority, a multisectoral VAW policy (including whether this includes the health sector), or clinical guidelines, by SDG regions and sub-regions.**
(DOCX)

**S3 Table. Proportion of countries with a national health policy that includes VAW as a strategic priority, a multisectoral VAW policy (including whether this includes the health sector), or clinical guidelines, by World Bank income groups.**
(DOCX)

**S4 Table. Proportion of countries that include clinical enquiry or universal screening in policy, by SDG regions and World Bank income groups.**
(DOCX)

**S5 Table. Proportion of countries that include first line support in policy, by SDG regions and World Bank income groups.**
(DOCX)

**S6 Table. Proportion of countries that include abortion for survivors of VAW in policy, by SDG regions and WB income groups.**
(DOCX)

**S7 Table. Proportion of countries that include mental health assessment and referral in policy, by SDG regions and World Bank income groups.**
(DOCX)

**S8 Table. Proportion of countries that include mental health treatment in policy, by SDG regions and World Bank income groups.**
(DOCX)

## Acknowledgments

Team members who contributed to data entry and extraction were Sophie Baumgartner, Mohamed Harby, Vanessa Maag, Sophie Morse, Noor El Nakib and Claire Veyriras. The

VAW Policy Database was developed by Svetlozar Mihaylov and Zvezdalina Dimitrova. Thanks are extended to an external reference group constituted for the technical review of the methodology and content of the report: Sabeen Afzal (Pakistan, Ministry of National Health Services, Regulation and Coordination), Lina Digolo (Kenya, the Prevention Collaborative) and Asha George (South Africa, University of Western Cape, School of Public Health). Gratitude is also extended to the following WHO colleagues for their contributions: Stephanie Burrows (Department of Social Determinants of Health), Theresa Diaz, Elizabeth Katwan and Marcus Stahlholfer (Department of Maternal, Newborn, Child and Adolescent Health and Ageing); Ian Askew, Svetlin Kolev, Antonella Lavelanet, Khurshed Nosirov and Soe Soe Thwin (SRH Department), all of whom gave input and support at various stages during the process of compiling and preparing the VAW Policy Database. This process also benefited from inputs and collaboration from the following WHO Regional Advisors who serve as VAW focal points: Britta Baer (WHO Regional Office for the Americas/Pan American Health Organization [PAHO]), Taiwo Oyelade (WHO Regional Office for Africa), Hala Sakr-ali and Anna-Rita Ronzoni (WHO Regional Office for the Eastern Mediterranean), Isabel Yordi and Aåsa Nihlen (WHO Regional Office for Europe), Anjana Bhushan (WHO Regional Office for South-East Asia), and Isabel Espinosa and Jaitra Sathyandran (WHO Regional Office for the Western Pacific).

## Author Contributions

**Conceptualization:** Eva Burke, Katherine Watson, Judy Gold, Claudia Garcia-Moreno, Avni Amin.

**Data curation:** Eva Burke, Katherine Watson, Gillian Eva, Judy Gold.

**Formal analysis:** Eva Burke, Katherine Watson, Gillian Eva.

**Methodology:** Eva Burke, Katherine Watson.

**Project administration:** Katherine Watson, Avni Amin.

**Validation:** Katherine Watson, Claudia Garcia-Moreno, Avni Amin.

**Writing – original draft:** Eva Burke.

**Writing – review & editing:** Eva Burke, Katherine Watson, Gillian Eva, Judy Gold, Claudia Garcia-Moreno, Avni Amin.

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
