## [Decision Letter · Decision Letter 0]

14 Dec 2023

Is addressing violence against women prioritised in health policies? Findings from a WHO Policies Database

PGPH-D-23-01845

Dear Ms Burke,

We are pleased to inform you that your manuscript 'Is addressing violence against women prioritised in health policies? Findings from a WHO Policies Database' has been provisionally accepted for publication in PLOS Global Public Health.

Best regards,

María De Jesús Medina Arellano, PhD

Academic Editor

Dear all, you have two excellent reviews, if possible please do try to address the comments that could make you improve your contribution, for example in what on the suggestions of reviewer one: ..."It would be interesting to know more data to understand the differences found by region, considering a more structural approach to the perception of violence, as well as the socio-cultural, economic and other aspects involved." If you don't think it is necessary, it will ok, but I believe it will give you further data to support your argument.

Reviewer Comments (if any, and for reference):

Reviewer's Responses to Questions

**Comments to the Author**

1. Does this manuscript meet PLOS Global Public Health’s publication criteria? Is the manuscript technically sound, and do the data support the conclusions? The manuscript must describe methodologically and ethically rigorous research with conclusions that are appropriately drawn based on the data presented.

Reviewer #1: Yes

Reviewer #2: Yes

2. Has the statistical analysis been performed appropriately and rigorously?

Reviewer #1: Yes

Reviewer #2: Yes

3. Have the authors made all data underlying the findings in their manuscript fully available (please refer to the Data Availability Statement at the start of the manuscript PDF file)?

Reviewer #1: Yes

Reviewer #2: No

4. Is the manuscript presented in an intelligible fashion and written in standard English?

Reviewer #1: Yes

Reviewer #2: Yes

5. Review Comments to the Author

Reviewer #1: It is a very good approximation of the situation of policies to eradicate violence against women, as it offers a broad panorama of their incorporation in the different countries and reveals the gaps in which the countries must work to achieve a comprehensive implementation with a favorable impact for women who have experienced violence.

Regarding the findings, it is very relevant to note in which countries the health sector does not play a preponderant role in joining public policies to eradicate violence against women, since different studies have pointed out that violence and its effects should be understood as a public health problem.

The emphasis on incorporating mental health care as a fundamental part of public policies and concrete actions against violence against women is very relevant, as well as the first-line support that people who have experienced violence should have in all cases.

Access to abortion as part of a comprehensive approach and as a right of victims is a fundamental fact that must be addressed by all countries, moving towards less restrictive regulatory frameworks, especially when abortion can be conceived as a harm reduction measure for many women victims of sexual violence.

Finally, the comparative analysis of the information suggests very significant elements for further work, because although for the moment it is a descriptive approach, in which the limitations were also expressed, in the discussion and conclusions section, future lines of research are shown. It would be interesting to know more data to understand the differences found by region, considering a more structural approach to the perception of violence, as well as the socio-cultural, economic and other aspects involved.

Reviewer #2: The contribution of this paper is significant to the field, as it rigorously analyzes information from a broad range of countries with an emphasis on gender and human rights.

The document enhances our understanding of how gender violence is addressed comprehensively in public health policies. The health sector plays a pivotal role in identifying and tackling some of the repercussions of gender-based violence.

Utilizing categories established by the World Bank to classify countries ensures that these findings can be compared with other studies and indicators. This approach will encourage further analysis by key stakeholders in the field and other researchers.

Given the research results, it might be beneficial to analyze the lack of accessibility and visibility of public policy documents related to this issue. The primary goal of adding briefly this analysis might be to highlight how this deficiency in dissemination and broad publication is affecting the implementation of these policies. If the individuals responsible for enacting them (e.i healthcare professionals, healthcare authorities, and so on) are unaware of their existence, their effectiveness could diminish. Additionally, the paper could benefit from a concise overview of the WHO’s primary recommendations for addressing gender violence within the healthcare sector.

I recommend its publication, as I believe it offers a valuable contribution to the field.

6. PLOS authors have the option to publish the peer review history of their article (what does this mean?). If published, this will include your full peer review and any attached files.

**Do you want your identity to be public for this peer review?** For information about this choice, including consent withdrawal, please see our Privacy Policy.

Reviewer #1: **Yes: **Gabriela Pineda Hernández

Reviewer #2: No
